# Super-Resolution Land Cover Mapping Based on the Convolutional Neural Network

**Yuanxin Jia** [1,2], **Yong Ge** [1,2,*], **Yuehong Chen** [3], **Sanping Li** [4], **Gerard B.M. Heuvelink** [5] and **Feng Ling** [6]

1 State Key Laboratory of Resources and Environmental Information System, Institute of Geographic Sciences and Natural Resources Research, Chinese Academy of Sciences, Beijing 100101, China
2 University of Chinese Academy of Sciences, Beijing 100049, China
3 School of Earth Sciences and Engineering, Hohai University, Nanjing 210098, China
4 DELLEMC CTO TRIGr, Beijing 100084, China
5 Soil Geography and Landscape Group, Wageningen University, P.O. Box 47, 6700 AA Wageningen, The Netherlands
6 Institute of Geodesy and Geophysics, Chinese Academy of Sciences, Wuhan 430077, China
* Correspondence: gey@lreis.ac.cn; Tel.: +86-10-6488-8053

**Abstract:** Super-resolution mapping (SRM) is used to obtain fine-scale land cover maps from coarse remote sensing images. Spatial attraction, geostatistics, and using prior geographic information are conventional approaches used to derive fine-scale land cover maps. As the convolutional neural network (CNN) has been shown to be effective in capturing the spatial characteristics of geographic objects and extrapolating calibrated methods to other study areas, it may be a useful approach to overcome limitations of current SRM methods. In this paper, a new SRM method based on the CNN (SRM$_{\text{CNN}}$) is proposed and tested. Specifically, an encoder-decoder CNN is used to model the nonlinear relationship between coarse remote sensing images and fine-scale land cover maps. Two real-image experiments were conducted to analyze the effectiveness of the proposed method. The results demonstrate that the overall accuracy of the proposed SRM$_{\text{CNN}}$ method was 3% to 5% higher than that of two existing SRM methods. Moreover, the proposed SRM$_{\text{CNN}}$ method was validated by visualizing output features and analyzing the performance of different geographic objects.

**Keywords:** super-resolution mapping; land cover; convolutional neural network; remote sensing imagery

## 1. Introduction

Land cover information is fundamental to many earth studies, such as natural resources management, urban planning, and land degradation analyses. Remote sensing (RS) is widely recognized as effective input for land cover mapping and change detection. In traditional land cover mapping, each pixel is assigned to a single class. However, because of the spectral resolution of RS images, the size and regular shape of pixels, and the heterogeneity of the earth's surface, it is inevitable that more than one land cover class appears in one pixel. The accuracy of land cover classification results estimated from RS imagery is confronted by challenges [1]. Spectral unmixing has been proposed to overcome the challenge of mixed pixels, and the proportional abundance of each class in a pixel can be estimated based on the spectral signatures of endmembers [2]. However, using these methods, the spatial position of each class within a pixel is still unknown. To obtain the spatial distribution of land cover in mixed pixels, super-resolution mapping (SRM) was proposed by Atkinson, and using this method a fine-scale spatial resolution land cover map can be achieved [3–6].

In the past two decades, a large number of SRM approaches and applications have been developed. Spatial dependence is often used as the basic assumption to derive the fine-scale land cover pattern. The within-pixel spatial location of land cover classes is determined by maximizing the spatial dependence between the classes in the subpixels and their neighboring pixels or subpixels [7–10]. The spatial dependence between a subpixel and pixels may be defined as the product of the inverse distances between each subpixel and its neighboring pixels of equal class, possibly using the class fraction values of the neighboring pixels as weights. Methods of this type include the subpixel/pixel spatial attraction model (SASPM) [11], learning-based algorithms [12], spatial interpolation methods (using radial basis functions), and vectorial boundary methods [3]. Several finer land cover results, which contain waterline and burned area, can be obtained by this kind SRM methods [13,14]. The spatial dependence between subpixels is calculated using an optimization strategy that switches the positions of subpixels until the spatial dependence of subpixels between neighboring pixels of the same type is maximized. This type of SRM solution is achieved by using numerical optimization methods, such as the Hopfield neural network [15–18], pixel swapping algorithm [19–21], maximizing posteriori method, genetic algorithm [14], and particle swarm optimization [4].

Although spatial dependence is a suitable strategy to simulate the distribution of the subpixel locations at fine-scale spatial resolution, spatial heterogeneity is common on the earth's surface, which may cause the spatial dependence of subpixels to be affected by irregular neighbors of different orientations [15,22]. Several SRM methods have been proposed to manage spatial heterogeneity. These SRM methods can be grouped into three categories. The first category considers prior information about the specific land surface structure, and solution approaches convert this prior information to spatial dependence constraints. The prior orientation of buildings, farmland, and object boundaries are extracted and considered as input or constraints for these SRM methods [22,23]. Furthermore, SRM can be performed separately for different types of geo-objects, and then combined in appropriate order [24]. The second category consists of geostatistical methods, which are used to manage spatial heterogeneity in different directions. The semi-variogram model between coarse fractional information and class probability of subpixel is derived and used to simulate the subpixel locations [25–27]. The third category makes use of auxiliary data. The auxiliary data can be categorized into finer images, multi-temporal difference coarse images and historic fine-scale land cover maps, and these are incorporated with coarse fractional information to estimate the fine-scale proportions of subpixels or used to directly swap subpixel positions [28–31].

For a homogeneous area with a simple and areal geo-object, spatial dependence-based SRM methods can obtain an accurate finer land cover map, and they have good robustness. Moreover, more heterogeneous information has be considered in the SRM methods when managing complex land surface. As mentioned in the above paragraph, results by SRM when combined with auxiliary data or information are excellent. More consideration needs to be focused on how to extrapolate to other areas, which can accelerate robustness and applicability of SRM for different geo-objects.

With the emergence of high-volume labeled data, high-performance computing, and state-of-the-art network structures, deep learning has demonstrated great advantages for image recognition. Specifically, the convolution neural network (CNN), which can automatically extract spatial features from image, has been shown to be successful for RS image applications, such as image classification, pixel classification and enhancement. RS scene classification is an important application of high-resolution spatial RS, where patches of an RS image are classified as exclusive classes. With the advantage of intrinsic feature extraction from RS patches, a CNN is suitable for scene classification. To overcome the shortage of samples when performing scene classification using a CNN, several sample expansion strategies have been adopted, such as ImageNet pre-trained models and self-labeling techniques [32]. Additionally, the learned features of different scales or nets can be combined to improve performance [33,34]. Another import application of a CNN for RS is pixel classification, which is commonly used for land cover or land use mapping from RS. The encoder-decoder CNN model is basic model, which is a down-sampled-then-up-sampled architecture [35], and multi-scale

features are extracted to maintain boundary information and reduce the categorical ambiguity [36–38]. Mohammadimanesh et al. [39] proposed a land cover mapping method from PolSAR data, where an Inception module and Residual module were adopted to extract more feature. Furthermore, several land cover mapping methods based on multi-model deep learning were proposed when using multi-temporal and multi-source data. Qiu et al. [40] proposed a land use mapping method from Sentinel 2 image, where the CNN was used for modelling the relationship between a multi-temporal Sentinel 2 images and corresponding land use classes. Interdonato et al. [41] proposed a dual view point deep learning model to map land cover from Sentinel 2 imagery, where a recurrent neural network and CNN were used to first extract temporal and spatial feature and combine these for classification. The third important application of the CNN in RS is image enhancement, where the CNN is used to construct a nonlinear relationship between the original and target image. Super-resolution construction is a typical example of this application, and several high spatial images have been derived from Landsat and Sentinel 2 images [42].

As mentioned above, the CNN has the advantage of extracting the intrinsic spatial features of RS images, and several CNN-based RS applications have achieved good performance. The ability of spatial feature extraction is suitable for SRM. Inspired by this, a CNN-based SRM method (SRM$_{CNN}$) is proposed in this paper. The main objectives of this research are as follows: (1) To propose a CNN-based SRM method; (2) to use feature visualization to illustrate how a CNN captures spatial features; (3) to analyze the performance of the proposed method for different geo-objects; and (4) to demonstrate the advantage of the CNN-based SRM method when compared with traditional methods. The remainder of this paper is organized as follows: In Section 2, the method is described. In Section 3, the details of data used are presented. The results are presented in Section 4. Several analyses are done in Section 5. Finally, the conclusions are stated in Section 6.

## 2. Methods

### 2.1. Basic Theory of Super-Resolution Mapping

The objective of SRM is to derive a fine-scale accurate land cover map from a coarse RS image, which means that the labels (i.e., land cover classes) of the subpixels within mixed pixels are determined by SRM. Suppose that the number of land cover types in a certain area ($N$ pixels) is $C$, then the fraction images of the $C$ land cover classes, represented as $X = (x_n^c | n = 1, \ldots, N; c = 1, \ldots, C)$ are first obtained by spectral unmixing or soft classification of the original coarse RS image. After the discrete zoom factor $z$ is set according to the targeted spatial resolution, each pixel is decomposed into $z^2$ subpixels, and the output finer land cover map obtained by SRM is represented as $Y = (y_{n,j}^c | n = 1, \ldots, N; j = 1, \ldots, z^2; c = 1, \ldots, C)$. All subpixels are regarded as pure pixels; this means that each $y_{n,j}^c$ is assigned the value 1 or 0, where 1 indicates that subpixels $y_{n,j}$ belong to class $c$ and 0 indicates that it belongs to another class. Spatial dependence $SD_{n,j}^c$ of a subpixel can be expressed by the attracting value of its neighboring pixels [11]. The optimal allocation of classes to the subpixels of mixed pixel is achieved by maximizing the spatial dependence between neighbor pixels under constraint that the class proportions within the mixed pixels are preserved [43]. Thus, the SRM method can be considered as spatial optimization approach [43]:

$$\text{maximize} \sum_{n=1}^{N} \sum_{c=1}^{C} \sum_{j=1}^{z^2} y_{n,j}^c \times SD_{n,j}^c \tag{1}$$

$$\text{subject to} \begin{cases} \sum_{c=1}^{C} y_{n,j}^c = 1 & \text{for all } n = 1, \ldots, N \\ \sum_{j=1}^{z^2} y_{n,j}^c = x_n^c \times z^2 & \text{for all } n = 1, \ldots, N \end{cases} \tag{2}$$

## 2.2. Background of the Convolutional Neural Network

The CNN is proposed for image recognition. A high correlation of local features and invariance to location shift of images are two fundamental properties of the CNN, and these are converted to a local connection, shared weights, pooling and iterative connection, which constitute the architecture of the CNN [44]. Traditional CNN structures contain a convolution (deconvolution) layer, pooling layer, and fully connected layer (or segment layer).

Convolutional layer: The convolutional layer is the most basic and important module in the CNN [45]. It is designed for feature extraction from images. The convolution layer contains a set of filters, which automatically extract specific features during the training stage. Each filter has dimensions $M \times M \times K$, where $M$ is the spatial size of the filter ($M$ is even normally), and $K$ is the channel number. For an original image or feature map of the previous layer with dimensions $W \times H \times K$, a convolution processing is conducted as following:

$$y_{m,n,\,k} = f\left( \sum_{i=m-\frac{M}{2}}^{m+\frac{M}{2}} \sum_{j=n-\frac{M}{2}}^{n+\frac{M}{2}} \sum_{k=1}^{K} x_{i,j,\,k} \times a_{ijk} + b_k \right), \tag{3}$$

where $x_{i,j,\,k}$ is the feature value from the original image or feature maps, $a_{ijk}$ is filter value, and $b_k$ is bias. Output $y_{m,n,\,k}$ is obtained by transformed the sum value via using an activation function $f(\cdot)$ [45]. The dimensions of $y_{m,n,\,k}$ depend on the number of filters, width of the receptive field and stride parameter $s$. Stride parameter $s$ is the pixel number between two consecutive convolution processing. The channel of output $y$ is equal to the number of filters. The receptive field represents the area of the input feature map to be convolved and stride represents the interval between two ordinal filters, the dimensions of the output $y$ are $\frac{(W-M+1)}{s} \times \frac{(H-M+1)}{s} \times K$.

The height and width of the output feature map decrease if the convolution layer becomes deeper. It then becomes less suitable for image segmentation (pixel classification). To maintain the output size, transposed convolutions (TransConv) or dilated convolutions have been proposed [46,47]. This means that a set of expansions or transformations is performed on the input feature map.

A nonlinear activation layer has been added to the convolution layer to overcome the problem of the vanishing gradient or overfitting. One of the most practical and commonly used activation layers is the rectified linear unit (ReLU), which is formulated as $x' = \max(0,\,x)$ [45].

Spatial pooling: The pooling layer has the ability to preserve spatial discriminant information and translation invariance [35]. Max pooling is the common strategy. It selects the highest value of a specific area as the new value of the next layer.

Classification layer: The last layer of a CNN is a classifier with differentiable loss, from which a class type or probability can be obtained. The common activation method is the softmax classifier as,

$$p(\hat{y}_i) = \frac{exp(x_i)}{\sum_{c=1}^{C} exp(x_c)}, \tag{4}$$

where $x_i$ is the C-dimensional input feature. Loss function is usually taken as the cross entropy [35],

$$L(y_i, \hat{y}_i) = \sum_{c=1}^{C} y_{ic} log \frac{1}{\hat{y}_{ic}} = - \sum_{c=1}^{C} y_{ic} log \hat{y}_{ic}. \tag{5}$$

## 2.3. Three Stages of the CNN for Super-Resolution Mapping

The main objectives of this research are to propose a CNN-based SRM method, open the "black box", and test its performance on different objects. Figure 1 shows a flowchart of the objectives and how they are achieved. It contains three main parts, which are the training stage, prediction stage, and analysis stage. In the training stage, a CNN model was adopted to simulate the relationship between

the coarse RS image and the fine-scale land cover map. Note that there are several studies in which a CNN was used to sharpen a coarse fraction image, and the obtained finer fraction image was used to derive the fine-scale land cover map, which is different from our research, in which the classic probability of each subpixel is directly obtained from the coarse RS image [48]. In the prediction stage, the learned CNN models from the training stage were adopted for coarse RS images to obtain class probabilities at the target finer resolution. The class probabilities were then converted to a land cover map. For the analysis stage, an accuracy assessment was performed to illustrate the performance of SRM$_{CNN}$, and feature visualization was achieved to explain the advantage of SRM$_{CNN}$.

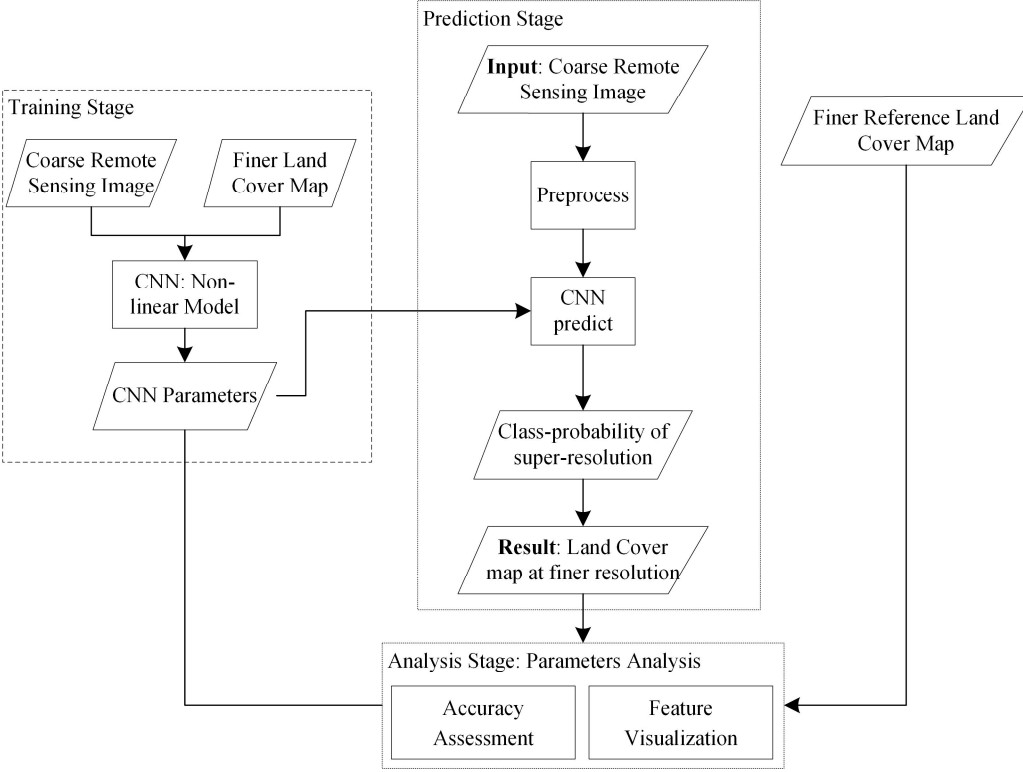

**Figure 1.** Three stages of flowchart.

### 2.4. Proposed SRM Network

The proposed SRM model (SRM$_{CNN}$) was inspired by state-of-the-art image super-resolution methods (ISR), such as super-resolution convolutional neural network [49], generative adversarial network for image super-resolution [50]. However, the major difference between ISR and SRM is that the pixel value in the former case is a spectral or luminance value, whereas, in case of the latter, it is a categorical or class value. In the proposed method, we adopted a similar-ISR model to obtain fine-scale land cover probabilities. Specifically, the last layer of ISR was replaced by a softmax classifier layer.

The model was inspired by enhanced deep super-resolution network (EDSR) [51], in which a multi-convolutional layer and skip connection were used to improve feature learning and solve the vanishing gradient problem. There were three main parts in the proposed network. The first part was a three-sequential convolutional layer with ReLU and pooling, which is shown in Figure 2. Each convolutional layer is composed of 64 filters, and the width and height of the receptive field are both set to 3. After the convolution procedure, the element-wise ReLU and max-pooling of $2 \times 2$ windows are performed. The second part is up-sampling, for which a multi transposed-convolutional layer was adopted. To keep the feature learned in the previous layer, a skip connection was used to concatenate the output of the corresponding convolution layer. The last part was the softmax classifier, in which

the feature in the antepenultimate layer was classified and class probabilities are obtained; detailed parameters of the model are shown in Table 1.

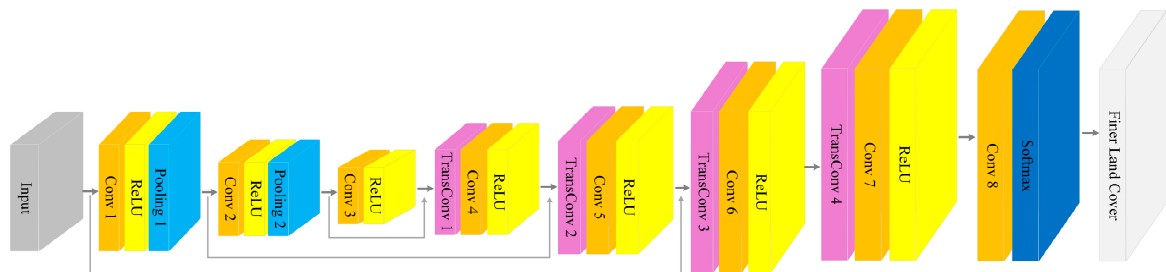

**Figure 2.** The proposed SRM$_{CNN}$ model.

**Table 1.** Parameters of the SRM$_{CNN}$ model.

| Layer | Filter Size | Filter Number | Stride | Pooling Window |
|---|---|---|---|---|
| Conv 1–3 | $3 \times 3$ | 64 | 1 | - |
| Conv 4–6 | $3 \times 3$ | 64 | 1 | - |
| Conv 7–8 | $3 \times 3$ | 32 | 1 | |
| TransConv 1–3 | $3 \times 3$ | 64 | 2 | - |
| TransConv 4 | $3 \times 3$ | 32 | 2 | |
| Pooling 1–2 | - | - | - | 2 |

## 3. Experiments

To evaluate the performance of the SRM$_{CNN}$ method, two experiments based on multi-spectral RS images were conducted. The descriptions of the dataset, training and testing procedure, and accuracy assessment metrics were presented below.

### 3.1. Datasets

The first dataset comprises public semantic labeling data from the International Society for Photogrammetry and Remote Sensing Commission (ISPRS) Vaihingen, Germany [52]. It contains 16 high resolution airborne images with corresponding fully labeled maps. The ground sampling distance of both data is 9 cm. The average size of the images is $2494 \times 2064$ and each image has four channels (near infrared, red, green, and digital surface models). Detailed information about the Vaihingen dataset is presented in Table 2. In order to meet the spectral unmixing requirement that the class number is not greater than the number of bands of the of RS image, only four classes were selected as the land cover map, which were *building*, *low vegetation*, *trees*, and *background* [1,52]. Fifteen images and their labeled maps were used to train the proposed SRM$_{CNN}$ net, and the remaining image was used to evaluate performance.

The second dataset is the Potsdam 2D Semantic Labeling dataset produced by ISPRS [52]. There are 25 fully labeled maps and the corresponding original multispectral images (red, blue, green and near infrared). The size of each image is $6000 \times 6000$ and the spatial resolution is 5 cm. Detailed information about the Potsdam dataset is presented in Table 2. In our experiments, a four-band image that had red, green, blue, and near infrared bands were used. Twenty four of the 25 images and corresponding labeled maps were used to train the proposed SRM$_{CNN}$ model, and one image was used to perform an accuracy assessment. The category of this dataset were *buildings, vegetation*, and *background*. The class number here was relatively small when compared with common land cover maps [53]. We used only three classes because in spectral unmixing the number of classes cannot exceed the number of bands and because Potsdam is a city with dense settlement buildings and has only a small number of different land cover types [52].

**Table 2.** Dataset parameters.

| Dataset | Bands | Spatial Resolution | Number | Average Size |
|---------|-------|-------------------|--------|--------------|
| Vaihingen | Red, Green, Near Infrared, DSMs | 9 cm | 16 | 2494 × 2064 |
| Potsdam | Red, Green, Blue, Near Infrared | 5 cm | 25 | |

### 3.2. Training and Prediction Procedure

A conventional training procedure was used for the proposed $SRM_{CNN}$, similar to EDSR, where weights and network bias were initialized randomly and optimized iteratively using stochastic gradient descent. Each gradient step consisted of a forward pass to compute the current loss over a small random batch of image patches, followed by back-propagation of the error signal through the network.

To generate the training dataset, which comprised the coarse image and fine-scale land cover pairs, the RS image was down-sampled to the desired ground spatial distance. Specifically, a Gaussian filter was adopted to blur the images with standard deviation $\sigma = 1/z$ pixel, where $z$ was the spatial resolution ratio between the coarse RS image and the finer land cover map [42]. The blurred images were sub-sampled with a target zoom factor. We used a zoom factor of 4, which mean that the ground spatial distance of the target land cover map was four times smaller than that of the coarse input image.

Considering the memory and computation limitations of the GPU, training images and land cover maps were split to small and discontinuous patches. The height and width of these patches were 32 × 32, and consequently these of the finer land cover map were 128 × 128. For the Vaihingen dataset, the 15 training images (down-sampled RS images and original land cover maps) were split into 4560 patches, and for Potsdam, the 24 images were split into 50,784 patches. The number of training and testing patches is presented in Table 3.

**Table 3.** Training and testing numbers.

| Dataset | | Image Number | Patches Number |
|---------|--|--------------|----------------|
| Vaihingen | Training | 15 | 4560 |
| | Test | 1 | - |
| Potsdam | Training | 24 | 50,784 |
| | Test | 1 | - |

In the prediction stage, the class probability of each subpixel (pixel of the super-resolution land cover map) was calculated by the proposed $SRM_{CNN}$ network with learned weights. To match the sizes of the patch and testing image, the prediction was performed in a sliding window of 32 × 32 with 1 stride of the test image.

The proposed network was implemented in the Keras framework [54], with TensorFlow as the backend. Training was run on an NVIDIA Titan Xp GPU, with 8 GB of RAM. The mini-batch size for SGD was set to 128 to fit into the GPU memory. The initial learning rate was $lr = 1 \times e^{-5}$ and reduced by a factor of 2 whenever the validation loss did not decrease for 5 consecutive epochs.

### 3.3. Baselines and Evaluation Metrics

The SRM methods of SASPM proposed by Mertens and VBSPM proposed by Ge [3,11] were used as the baseline methods. The inputs of both SASPM and VBSPM are coarse class fractional images (CFIs), which can be obtained using a soft classification or spectral unmixing method [43,55]. It should be noted here that the inputs of the proposed $SRM_{CNN}$ and baseline are different, which are the coarse RS image and fraction image, respectively. We address this issue in the discussion. To simulate the traditional SRM procedure, in which a fractional image was obtained from a coarse RS image, a CFI image was obtained using SVM from a down-sampled coarse image, and land cover maps at the target finer resolution were obtained using SASPM and VBSPM.

To measure and compare the performance of the SRM$_{CNN}$ method and baselines, the producer accuracy (PA) and user accuracy (UA) of each class, and the overall accuracy (OA) of test images were computed. PA and UA evaluate class-specific accuracies, where PA is the ratio of the correctly classified area to the entire area for a specific class and UA is the same ratio but calculated on the reference map. Reference maps were taken as the entire semantic land cover map from the dataset.

## 4. Results

In the prediction stage, the original high-resolution RS images from the Vaihingen and Potsdam datasets were first blurred using a Gaussian filter and then down-sampled by a factor of four. The down-sampled images (coarse spatial resolution) were then submitted to the baseline SRM methods and proposed SRM$_{CNN}$ method. For the baseline SRM methods, fraction images were first obtained using an SVM soft classification method with selected samples, and a fine-scale land cover map was created using VBSPM and SASPM. For the SRM$_{CNN}$ method, a class-affiliation probability at a fine-scale spatial resolution was created directly using the learned weights, and a fine-scale land cover map was obtained by selecting the class with the highest probability.

### 4.1. Results for Vaihingen Dataset

The original image of "area34" was selected as a test image for the Vaihingen dataset because it was sufficiently far from the training areas. The fine-scale land cover maps obtained by SASPM, VBSPM and SRM$_{CNN}$ are shown in Figure 3. An accuracy assessment for each map was performed by comparing entire pixels of resulting maps with the reference land cover map. The confusion matrices of all the methods are presented in Tables 4–6.

Figure 3 clearly shows that the spatial distribution of the land cover maps from SASPM, VBSPM, and SRM$_{CNN}$ were similar to those of the reference map, where *building* locates along with *background* and was surrounded by *low vegetation* and *tree*. Generally, the SRM$_{CNN}$ result was more similar to the reference map when compared with SASPM and VBSPM. However, the detailed spatial distribution in several subareas was obviously different from that of the reference map, particularly for *low vegetation* and *tree*.

The accuracy assessments of the three simulated land cover maps confirmed the aforementioned findings. The OAs of the three results were all above 77%, which demonstrates the good fitness between the simulated results and reference map. The OA of SRM$_{CNN}$ surpassed that of the two baseline methods. It was 5% and 6% higher than that of VBSPM and SASPM, respectively.

Detailed information about the class-specific accuracy supports the above results. The PA and UA of *low vegetation* were relatively lower than those of other classes, the average class-specific accuracy of *low vegetation* was approximately 0.62. The main reason for the relatively low accuracy was that *low vegetation* was often misclassified as *tree*. The average misclassification was 23% for PA and 15% for UA. The *background* class had the best performance, with an average PA of 0.92. Average PA of *building* and *tree* class were 85% and 84% respectively, which showed a good performance.

It could be concluded from the accuracy assessment that SRM$_{CNN}$ performed better than SASPM or VBSPM for most class-specific accuracies. A detailed comparison of the three SRM methods was shown in the two zoom-in areas in the second and third rows of Figure 3. In the zoom-in areas, the SRM$_{CNN}$ image had less "salt–pepper" noise than the SASPM and VBSPM images. Moreover, the spatial distribution of the SRM$_{CNN}$ image agrees more with the reference map.

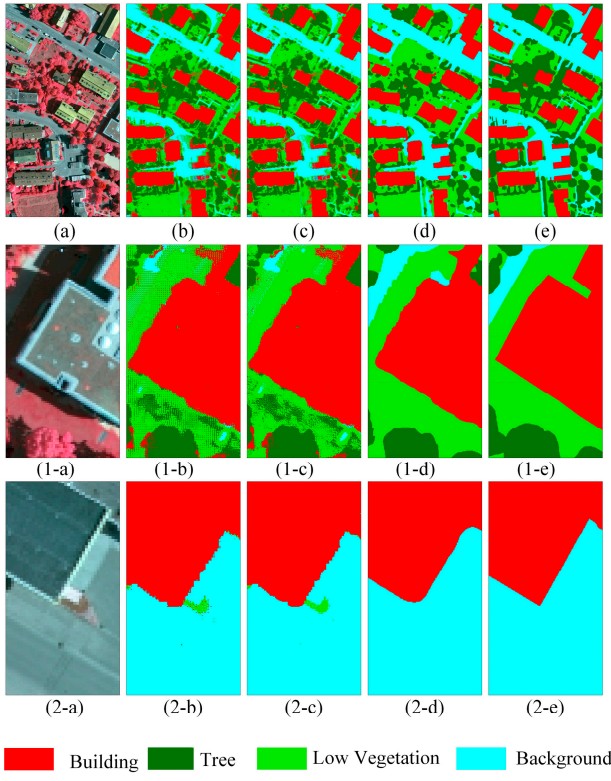

**Figure 3.** Results of Vahingen Dataset. (**a**) Coarse image; (**b**) SASPM result; (**c**) VBSPM result; (**d**) SRM$_{CNN}$ result; and (**e**) reference map. The second and third row were zoom-in areas from the first row.

**Table 4.** Confusion matrix of SASPM result for Vahingen Dataset.

|  |  | Reference |  |  |  |  |
|---|---|---|---|---|---|---|
|  |  | Background | Building | Low Vegetation | Tree | PA |
| Result | Background | 593,604 | 12,242 | 18,390 | 4320 | 0.92 |
|  | Building | 48,490 | 862,340 | 50,310 | 86,752 | 0.82 |
|  | Low Vegetation | 143,482 | 18,525 | 586,454 | 204,855 | 0.61 |
|  | Tree | 4651 | 9644 | 175,597 | 696,139 | 0.79 |
|  | UA | 0.75 | 0.96 | 0.71 | 0.70 | OA = 0.77 |

**Table 5.** Confusion matrix of VBSPM result for Vahingen Dataset.

|  |  | Reference |  |  |  |  |
|---|---|---|---|---|---|---|
|  |  | Background | Building | Low Vegetation | Tree | PA |
| Result | Background | 590,096 | 10,945 | 14,484 | 3015 | 0.93 |
|  | Building | 40,376 | 865,377 | 46,417 | 81,770 | 0.84 |
|  | Low Vegetation | 156,955 | 17,597 | 597,862 | 207,619 | 0.60 |
|  | Tree | 1977 | 8832 | 171,934 | 699,657 | 0.79 |
|  | UA | 0.75 | 0.96 | 0.72 | 0.71 | OA = 0.78 |

**Table 6.** Confusion matrix of SRM$_{CNN}$ result for Vahingen Dataset.

|  |  | Reference |  |  |  |  |
|---|---|---|---|---|---|---|
|  |  | Background | Building | Low Vegetation | Tree | PA |
| Result | Background | 689,600 | 17,100 | 40,429 | 19,528 | 0.89 |
|  | Building | 12,303 | 840,268 | 13,162 | 19,483 | 0.95 |
|  | Low Vegetation | 68,967 | 29,091 | 728,624 | 320,821 | 0.63 |
|  | Tree | 3852 | 5846 | 32,842 | 623,789 | 0.94 |
|  | UA | 0.87 | 0.93 | 0.88 | 0.63 | OA = 0.83 |

### 4.2. Result of Potsdam Dataset

The test image for the Potsdam dataset was "area_7_11". The fine-scale land cover maps and accuracy assessment of SASPM, VBSPM, and SRM$_{CNN}$ were obtained using the same approach used for the Vaihingen dataset. The simulated fine-scale land cover and accuracy assessment results were shown in Figure 4 and Tables 7–9.

Figure 4 shows that *building* was mainly located in the left-upper part of the area and was separated from a main road (*background*). Generally, all classes of the simulated land cover maps of SASPM, VBSPM and SRM$_{CNN}$ had a similar structure to the reference map. Few differences occurred between the three simulated results.

Although the OA of the three simulated results was greater than 80%, the result of SRM$_{CNN}$ improved slightly compared with the other methods. It had a 2% and 3% higher OA than VBSPM and SASPM, respectively. Moreover, the class-specific accuracy of SRM$_{CNN}$ was always higher than those of SASPM and VBSPM, which demonstrates the ability of SRM$_{CNN}$ to better model the nonlinear relationship between the coarse image sensing image and the fine-scale land cover map.

Almost all class-specific accuracies of the three simulated results were higher than 80%, except for PA of *building*. The details of the confusion matrices of the three results showed that the pixels misclassified as *background* contributed to the low accuracy. However, PA of *building* was the highest among the three classes, and equaled 91%. PA of *vegetation* and *background* were almost equal, which was 82%.

Similar to the Vaihingen dataset, a detailed comparison of the three SRM methods was shown in two zoom-in areas in the second and third rows of Figure 4. In the zoom-in area, the SRM$_{CNN}$ image had less "salt–pepper" noise than the SASPM and VBSPM images. Moreover, the spatial distribution of the SRM$_{CNN}$ result was more similar to that of the reference map.

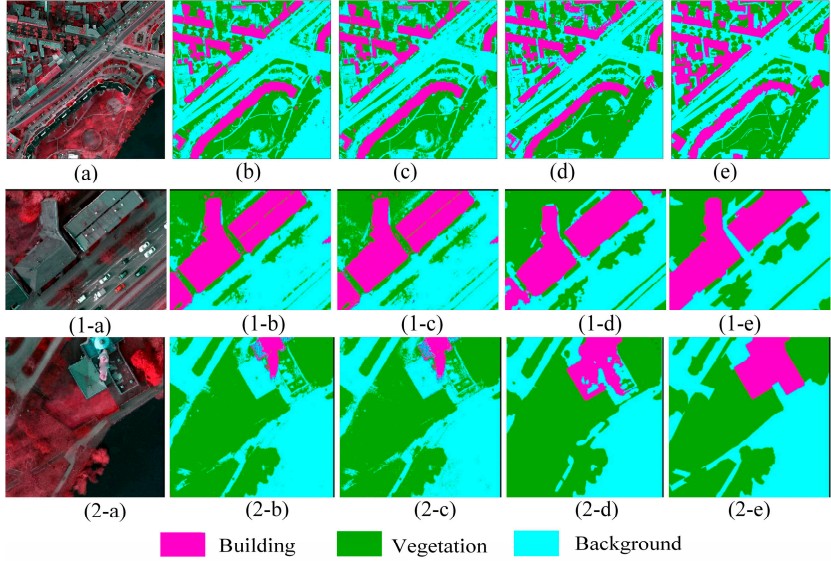

**Figure 4.** Results of Potsdam dataset. (**a**) Coarse image; (**b**) SASPM result; (**c**) VBSPM result; (**d**) SRM$_{CNN}$ result; and (**e**) reference. The second and third row were zoom-in areas from the first row.

**Table 7.** Confusion matrix of SASPM result for Potsdam dataset.

| | | Reference | | | |
| --- | --- | --- | --- | --- | --- |
| | | **Background** | **Building** | **Vegetation** | **PA** |
| Result | Background | 13,920,870 | 1,741,689 | 1,711,104 | 0.80 |
| | Building | 558,257 | 5,412,002 | 117,220 | 0.89 |
| | Vegetation | 1,829,913 | 704,592 | 10,004,353 | 0.80 |
| | UA | 0.85 | 0.69 | 0.85 | OA = 0.81 |

**Table 8.** Confusion matrix of VBSPM result for Potsdam dataset.

|  |  | Reference | | | |
|  |  | Background | Building | Vegetation | PA |
|---|---|---|---|---|---|
|  | Background | 13,940,867 | 1,748,038 | 1,678,244 | 0.80 |
| Result | Building | 559,091 | 5,423,543 | 111,263 | 0.89 |
|  | Vegetation | 1,809,082 | 686,702 | 10,043,170 | 0.80 |
|  | UA | 0.85 | 0.69 | 0.85 | OA = 0.82 |

**Table 9.** Confusion matrix of SRM$_{CNN}$ result for Potsdam dataset.

|  |  | Reference | | | |
|  |  | Background | Building | Vegetation | PA |
|---|---|---|---|---|---|
|  | Background | 13,800,985 | 1,760,361 | 706,099 | 0.85 |
| Result | Building | 236,141 | 5,832,048 | 33,361 | 0.96 |
|  | Vegetation | 2,271,914 | 265,874 | 11,093,217 | 0.81 |
|  | UA | 0.85 | 0.74 | 0.94 | OA = 0.85 |

## 5. Discussion

### 5.1. The Advantage of CNN for Super Resolution Mapping

As shown in Tables 4–9 and in Figures 3 and 4, the SRM$_{CNN}$ had a better performance when compared with the baseline methods on both qualitative and quantitative analyses. From visually checking, the finer land cover map produced by SRM$_{CNN}$ had less noise (i.e., "salt and pepper"). From sub-plots of the second and third rows of Figure 3, the boundary between *building* and *low vegetation* or *background* by SRM$_{CNN}$ agreed better with the reference map than baseline methods. Additionally, the shape of the finer land map units as derived using SRM$_{CNN}$ was closer to what experts expect. An obvious phenomenon could be found in subplots from Figure 4, where the green belt in the bottom-right corner was recognized by SRM$_{CNN}$, but missed by the baseline methods. Quantitative analyses showed that most class-specific accuracies by SRM$_{CNN}$ were higher than those from the baseline methods for the Vahingen dataset, except for UA of *tree* and PA of *building*. PA of *building* and *tree*, UA of *background* and *low vegetation* from the Vahingen dataset were significantly improved by SRM$_{CNN}$, where the improvement could be 10% comparing with SASPM or VBSPM. Furthermore, all class-specific accuracies of the Potsdam dataset by SRM$_{CNN}$ were higher than those of the baseline methods. The average improvement was 4.5%.

The advantage of SRM$_{CNN}$ in visual and quantitative accuracy assessment was shown in the paragraph above, which indicated that SRM$_{CNN}$ had the ability to cope with traditional problems of common baseline method. The spectral characteristic of *background* and *building* class was similar, which means that it could be difficult to distinguish them using a common spectral-based method. It was clear from Tables 4 and 5 (or Tables 7 and 8) that misclassification between *building* and *background* was one of the main reasons for the relative low PA for SASPM and VBSPM. However, this problem had been well handled by SRM$_{CNN}$, as it had the ability to extract more spatial features.

### 5.2. Reason for SRM$_{CNN}$ Outperforms Baselines

In seeking to understand how the proposed SRM$_{CNN}$ network extracted features and why it performed better than other state-of-the-art SRM methods, an in-depth feature visualization was conducted. For clarity, only some parts of the layers were visualized. In particular, output features of the first, ninth, and 18th layers for a specific zone are shown, which were typical examples of feature extraction, higher feature fusion, and decoder (super-resolution). Specifically, the first, ninth, and 18th layers represented the output of Conv 1, TransConv 1 and TransConv 4 of Figure 2, respectively. To clarify the ID of the feature maps, feature$_{i,j}$ with subscript was used to represent the output feature of a specific row and column, where feature$_{1,1}$ was the upper-left corner cell.

As shown in Figure 5d, the output features of the first layer demonstrated that the coarse edge information of different types of geo-objects could be extracted. Specifically, several features, such as feature$_{1,1}$ and feature$_{5,4}$, were responsive to the *vegetation* class of the coarse RS image, and other features (e.g., feature$_{1,7}$) were learned to simulate the *building* class. However, the features of this layer had some drawbacks: One was that the boundaries between classes are unclear and another was that several classes cannot be distinguished.

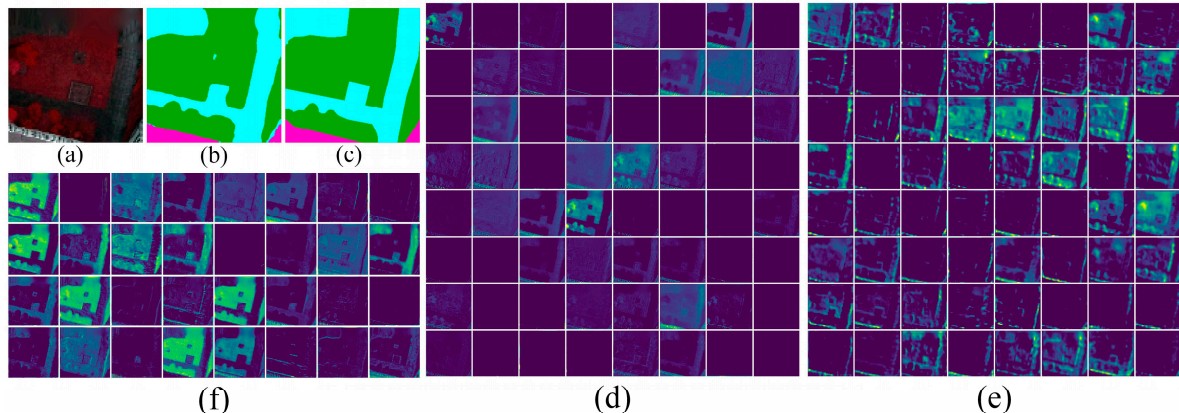

**Figure 5.** Features visualization. (**a**) Input coarse remote sensing images; (**b**) simulated result; (**c**) reference result; (**d**) visualization of first output features; (**e**) visualization of ninth output features; (**f**) visualization of 18th output features.

Unlike the first layer, more abstract features occurred in the ninth layers, as shown in Figure 5e. Edge information between different classes was becoming clearer. As shown in the bottom-left part of feature$_{6,5}$ and the upper right part of feature$_{7,6}$, the edge between *building* and *vegetation* could be recognized. Additionally, the characteristics of different classes were shown, such as feature$_{1,7}$, which was representative of *vegetation*, and feature$_{4,1}$, which was representative of *background*. Although edge information and characteristics could be seen in this layer, the boundary between different classes was not clear.

Figure 5f showed the output feature of the 18th layer. It could be seen from this layer that the boundaries and characteristics between classes were distinct. Feature$_{1,1}$, feature$_{2,1}$, and feature$_{3,5}$ were highly correlated with *vegetation*; feature$_{2,4}$ and feature$_{2,8}$ represented *background*; whereas feature$_{3,6}$ and feature$_{4,8}$ were representative of the *building* class.

To evaluate the performance of SRM$_{CNN}$ for different geo-objects, the test image, predicted result and reference were clipped into different patches. The accuracy metrics used in the results section (PA, UA, and OA) were computed for each patch. Several patches with high OA but low class-specific accuracy were selected for illustration. Reasons for the above selection strategy could be grouped into two aspects. Firstly, high OA mean that there was few noises in the patches, which might reduce uncertainty from data and focus on the effect of SRM$_{CNN}$. Secondly, lower class-specific accuracy mean that several geo-objects could not be recognized by SRM$_{CNN}$, which brought new orientation for SRM methods when using the CNN.

The selected three patches were shown in Figure 6. It could be seen that the mapping result of several areal geo-objects was accurate, such as the vegetation of three patches, where both the shape and boundary were similar to those of the reference. However, a normal phenomenon was that the predicted results of SRM$_{CNN}$ missed or added several small objects, such as the *building* class in Figure 6(2-b,2-c). The cause of this deficiency was that the proposed SRM$_{CNN}$ had several limitations in terms of extracting features of inconsistently sized geo-objects.

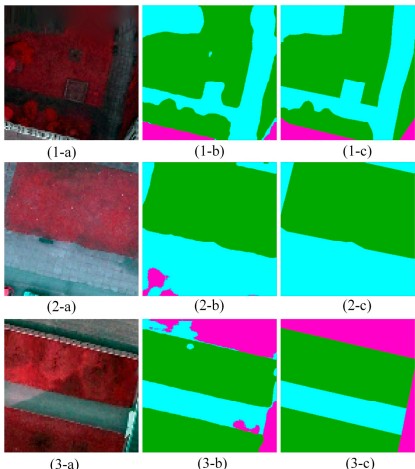

**Figure 6.** Examples of the performance of SRM$_{CNN}$ for different geo-objects: 1, 2, and 3 represent the patch number; and a, b, and c are the input coarse remote sensing images, predicted result, and reference, respectively.

### 5.3. Implications for Future Work

In this paper, the CNN had been tested for super resolution land cover mapping, and it had shown to perform better than existing methods both in visual and quantitative accuracy assessment. However, the number of land cover classes used in this paper was relatively small [53,56]. In future work it is important to evaluate the applicability of the CNN for super-resolution land cover mapping from Landsat or Sentinel 2 imagery. Fortunately, several CNN-based super reconstruction methods have been proposed for RGB, Sentinel 2, and Landsat imagery across the world [42,57,58], and these networks can be practical examples for super resolution mapping. Additionally, the current land cover maps or samples from Landsat or Sentinel 2 can be converted into training samples [59,60].

It should be noted that inputs of conventional SRM methods and the proposed SRM$_{CNN}$ were different. The input of former was the fraction image, which could be obtained from spectral unmixing of soft classification method; but that of the latter was a coarse remote sensing multiband image. Although the proposed SRM$_{CNN}$ method outperformed conventional SRM methods, the uncertainty caused by different input was needed to be discussed in future work. Although spectral unmixing methods have made progress, uncertainty still occurs in the state-of-the-art spectral unmixing methods [61], which will still affect the baseline SRM methods. Decreasing this uncertainty is one way to compare CNN-based SRM methods and conventional SRM methods. One solution is to use the simulated fraction image, where the uncertainty of unmixing can be eliminated. Another solution is that the fraction image is adopted as input for both SRM$_{CNN}$ and baseline SRM methods by state-of-the-art spectral-unmixing methods, from which an accurate finer land cover map can be obtained.

Although SRM$_{CNN}$ outperformed common state-of-the-art SRM methods, the results were not perfect and still had several shortcomings. The edge part between two classes was still mistaken, and some parts of tree was missed. The reason for this phenomenon was that not all features in the encoder-decoder network were useful for edge detection, and even worse that several features might confuse the network for recognition [62]. Multi-scale information has been tested as a way forward to tackle these issues. Natural advantage for remote sensing society is that a series of multi resolution images (such as Landsat 8 OLI, Sentinel 2 MSI, and SPOT 7) can be used for multi-scale feature extraction.

### 6. Conclusions

Inspired by the success of the CNN, when dealing with image classification, segmentation, and super resolution construction, a CNN-based SRM method (SRM$_{CNN}$) was proposed in this paper. An

encoder-decoder CNN network was used to simulate the nonlinear relationship between a coarse image and a fine-scale land cover map. Two experiments were conducted, using the Vaihingen and Potsdam datasets, to compare $SRM_{CNN}$ with baseline SRM methods. The results demonstrated that $SRM_{CNN}$ achieved an improvement of 5% or 6% for the OA on the Vaihingen dataset, and 2% or 3% on the Potsdam dataset. In addition to the accuracy improvement, visual checks showed that $SRM_{CNN}$ was more similar to the reference map when compared with SASPM and VBSPM.

Despite $SRM_{CNN}$ showing advantages, several deficiencies occurred and limited its applicability for real-world finer land cover mapping. In order to cope with these issues and boost the $SRM_{CNN}$ applicability, future work is needed on networks selection, training sample collections, and multi-scale integration.

**Author Contributions:** Y.J., Y.G., S.L., Y.C., G.B.M.H. and F.L. conceived the main idea and designed and performed the experiments. The manuscript was written by Y.J. and improved by contributions of all co-authors.

**Funding:** This work was supported by the National Natural Science Foundation for Distinguished Young Scholars of China (Grant No. 41725006), and two Key Programs of the National Science Foundation of China (Grant No. 41531174 and No. 41531179)

**Acknowledgments:** The dataset in this paper was obtained from International Society for Photogrammetry and Remote sensing (http://www2.isprs.org/commissions/comm3/wg4/semantic-labeling.html), and the Keras framework was downloaded from https://github.com/keras-team/keras. We thank all the contributors for the dataset and framework.

**Conflicts of Interest:** The authors declare no conflicts of interest.

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
