# Peer review of "Super-Resolution Land Cover Mapping Based on the Convolutional Neural Network"

_remotesensing, doi:10.3390/rs11151815_

Round 1

Reviewer 1 Report

nicely presented work, would be nice to be tested on a completely different image source.

Author Response

Thank you for your hard and constructive comments. We think we will test the work for real finer land cover mapping in future work.

Reviewer 2 Report

This paper proposes an innovative and interesting finding, which is a possibility of deep learning for super resolution mapping of landcover. However, descriptions are sometimes difficult for readers to correctly understand, partly because of mistaken and/or irregular use of words and symbols. In fact, I could not read entire manuscript because of these troubles. I suggest the authors to check and revise the manuscript carefully again submit it as a new submission.

Here are some places which I felt difficult to understand. Please remember that there should be similar problems in other places also.

1. Wrong use of symbols denoting equations and variables. Please refer to SI official document such as:
https://www.nist.gov/document/special-publication-330
In this document, policy of choice of symbols (italic or roman, bold ornon-bold, etc.) are clearly specified. It is an international agreement in science. For example, use of italic letters for writing exp or log is wrong. It must be in roman. Using bold in wrinting n=1, ..., N is also strange.

2. Inconsistent use of symbols. For example, K is written in roman in line 136 but is written in italic in line 136. SD_j^c in equation (1) is the same as SD_j^c in line 126 but they are written in different style.

3. Perhaps sum (sigma symbol) from n=1 to N is necessary in right side of equation (2).

4. { and } in line 120 and 121 must be ( and ), so as to make it consistent with line 117.

5. What is definition of "spatial dependence"? (line 126)

6. Line 140, 158, 160. Perhaps better to give them equation numbers.

7. There must be connection between p, q and m, n in the equation in line 140.

8. Line 158. p(y|x_i) never appears after this. Perhaps it is the same as y-hat_i in line 160. If it is true, it's better to use y-hat_i instead of p(y|x_i).

9. Line 158. It is better to mention that it is a softmax, because you mention softmax in the later text.

10. Figures contain letters and symbols which are too small to read. For example, Conv, ReLU, Pooling in Figure 2 are too small. 1-a, 1-b, ... in Figure 6 are also too small.

11. What is 32 filter bank? (line 188)

12. Is the "receptive field" in line 188 same as "Filter" in Table 1?

13. Table 1 is not consistent in Figure 2. For example, there is no "Conv1" in Figure 2.

14. "fully labeled map" in line 207 and "fully labelled maps" in line 214. Why only "s" in the line 207 and not in 214?

15. You said "16 RS images" in line 206 but only 15 in "Number" in Table 2. I am confused because they must be the same.

16. Perhaps "remaining" rather than "remain" in line 219.

17. What is z in line 232?

18. sigma is better than delta, perhaps (line 232).

19. I do not understand line 254-259 well. Does "coarse image" in line 258 mean a RS image or a landcover map? From line 256, I guessed that you actually did a soft classification or spectral unmixing, but I am confused because in other places it sounds like you took only coarse landcover map as an input to the baseline methods.

20. I desire discussion about mistaken result of Figure 3. For example, some parts of Tree in reference data are somehow reproduced in SASPM and UBPSM but not in SRM_CNN.

21. Same suggestion for Figure 4.

22. I do not understand "1st, 9th, and 18th layers" in Figure 5. Where do these layers locate in Figure 2?

23. Index of "feature" is confusing and somewhat mistake. In line 349, you said "feature_i, j with subscripts is ... column and row". I interpret that i is column and j is row. But in line 363 and 364, it looks like i is row and j is column.

24. Line 364, mention about feature_7,6 ... I do not agree. I cannot see the edge.

25. line 138. k_th filter must be k-th filter, perhaps.

26. line 138. k_ijk is strange. "k" appears twice in different meaning.

Author Response

This paper proposes an innovative and interesting finding, which is a possibility of deep learning for super resolution mapping of landcover. However, descriptions are sometimes difficult for readers to correctly understand, partly because of mistaken and/or irregular use of words and symbols. In fact, I could not read entire manuscript because of these troubles. I suggest the authors to check and revise the manuscript carefully again submit it as a new submission.

Here are some places which I felt difficult to understand. Please remember that there should be similar problems in other places also.

1. Wrong use of symbols denoting equations and variables. Please refer to SI official document such as: https://www.nist.gov/document/special-publication-330

In this document, policy of choice of symbols (italic or roman, bold ornon-bold, etc.) are clearly specified. It is an international agreement in science. For example, use of italic letters for writing exp or log is wrong. It must be in roman. Using bold in wrinting n=1, ..., N is also strange.

Response:

Thank you very much for your constructive comments. We have checked the whole paper, and all symbols and equations have been checked on the correct use of SI notation. We have also corrected the wrong use of italics and bold.

2. Inconsistent use of symbols. For example, K is written in roman in line 136 but is written in italic in line 136. SD_j^c in equation (1) is the same as SD_j^c in line 126 but they are written in different style.

Response:

Thank you very much for spotting these inconsistencies. We have revised the notation and equation, see which are shown in line 133 and Equation 1 and 2 in the new manuscript. We have also checked and corrected inconsistencies in the use of symbols of the entire manuscript.

3. Perhaps sum (sigma symbol) from n=1 to N is necessary in right side of equation (2).

Response:

The constraints must be valid for all n=1,…,N and we have now made this clear in Equation 2. We have summed over all N mixed pixels in Equation 1, to make clear that the optimization is done jointly for all mixed pixels in the study area.

4. { and } in line 120 and 121 must be ( and ), so as to make it consistent with line 117.

Response:

We have made it consistent and now use ( and ) in both cases. See lines 127 and line 131.

5. What is definition of “spatial dependence”? (line 126

Response:

We have added a sentence to describe spatial dependence and provided a reference, see which is shown in lines 133-134.

6. Line 140, 158, 160. Perhaps better to give them equation numbers

Response:

We have added equation numbers.

7. There must be connection between p, q and m, n in the equation in line 140.

Response:

We apologize for this inconsistency and corrected the use of symbols. The new equation (now on line 151) no longer has symbols p and q.

8. Line 158. P(y|x_i) never appears after this. Perhaps it is the same as y-hat_i in line 160. If it is true, it’s better to use y-hat_i instead of p(y|x_i).

Response:

We agree that this is better and have made the adjustment. Thank you.

9. Line 158. It is better to mention that it is a softmax, because you mention softmax in the later text.

Response:

We mention that it is the softmax function just before we present the equation (lines 171-172).

10. Figures contain letters and symbols which are too small to read. For example, Conv, ReLU, Pooling in Figure 2 are too small. 1-a, 1-b, ... in Figure 6 are also too small.

Response:

We have revised Figures and Tables and increased the font size.

11. What is 32 filter bank? (line 188)

Response:

We corrected this and replaced ‘filter banks’ with ‘filters’.

12. Is the "receptive field" in line 188 same as "Filter" in Table 1?

Response:

We have rephrased this sentence and removed receptive field, which is shown in line 148.

13. Table 1 is not consistent in Figure 2. For example, there is no "Conv1" in Figure 2.

Response:

Thank you for spotting this error. We have corrected it and made it consistent.

14. "fully labeled map" in line 207 and "fully labelled maps" in line 214. Why only "s" in the line 207 and not in 214?

Response:

We have checked the whole paper, and changed them to “fully labeled maps”.

15. You said "16 RS images" in line 206 but only 15 in "Number" in Table 2. I am confused because they must be the same.

Response:

It’s a mistake, we have changed 15 to 16, which is shown in Table 2.

16. Perhaps "remaining" rather than "remain" in line 219.

Response:

Thank you for your spotting this mistake, we rephrased the text.

17. What is z in line 232?

Response:

We have added a sentence and a reference to describe it, as which is shown in lines 251-252.

18. sigma is better than delta, perhaps (line 232).

Response:

We have revised it.

19. I do not understand line 254-259 well. Does "coarse image" in line 258 mean a RS image or a landcover map? From line 256, I guessed that you actually did a soft classification or spectral unmixing, but I am confused because in other places it sounds like you took only coarse landcover map as an input to the baseline methods.

Response:

We agree that we were not very clear. In order to clarify this issue, we have added several sentences. See lines 277-278. We also address it in the Discussion part, see lines 451-461.

20. I desire discussion about mistaken result of Figure 3. For example, some parts of Tree in reference data are somehow reproduced in SASPM and UBPSM but not in SRM_CNN.

Response:

Several shortcomings still occur for the result of proposed method, e.g. the edge information and tree was mistaken. We have searched several relative reference papers, and have give some solution for these issues in future work. See lines 463-469.

21. Same suggestion for Figure 4.

Response:

Same with above.

22. I do not understand "1st, 9th, and 18th layers" in Figure 5. Where do these layers locate in Figure 2?

Response:

These numbered layers correspond to the sequence of layers in Figure 2. For example, the 1st layer represents the first Conv layer. We agree that this was unclear and have inserted a sentence that explains what these refer to (lines 397-398).

23. Index of "feature" is confusing and somewhat mistake. In line 349, you said "feature_i, j with subscripts is ... column and row". I interpret that i is column and j is row. But in line 363 and 364, it looks like i is row and j is column.

Response:

We re-arranged the order and now write “row and column”. See lines 399-400.

24. Line 364, mention about feature_7,6 ... I do not agree. I cannot see the edge. 

Response:

It’s a very small edge, we have made it clearer. We also explained where the edges are within the square cell (lines 413, 414)

25. line 138. k_th filter must be k-th filter, perhaps.

Response:

We rephrased the text and no longer use kth filter

26. line 138. k_ijk is strange. "k" appears twice in different meaning.

Response:

We agree and replaced one ‘k’ with ‘a’. See lines 151-152.

Reviewer 3 Report

In this manuscript authors analyze CNN methods to obtain fine-scale land cover maps. It is a well written manuscript, however authors have to provide more information, references, etc.

Introduction: I would like to have more information about how theses techniques have been tested and used to generate land cover maps.

Methods: You write absolute sentences without any reference supporting your writing. For example, line 133; "The convolution layer is the most basic and inportant module in CNN", is it your opinion or are there evidences to write it?. Review your document.

Figures are difficult to read, for example, in Fig 1 and 2 text is too small.

3.1. Datasets: Please provide references as web pages for datasets used. Moreover, can you provide information about image characteristics. I mean, sensor, pixel size...

Results: I appreciate if you show add producer accuracy. It would be interesting a deep analyze between classes. What is background class? 

You have worked with just only 3 classes. What is the effect of number of classes? A real Land cover maps is not only 3 classes.

Dicussion section: You do not discuss anything at all, you just show results. Please rewrite it, link with previous results from other researcher results.

Conclusion: Please, rewrite it. It is a very short summary of your result section.

Finally, what about future research? 

How can your methods can be incorporated in real flowchart to generate land cover maps?

Yo have used fine spatial resolution with 9 and 5 cm, what pixel size effect?. I mean, what is your opinion about this methodology if a used Landsat or Sentinel 2 images to generate a land cover map of a country? 

Author Response

In this manuscript authors analyze CNN methods to obtain fine-scale land cover maps. It is a well written manuscript, however authors have to provide more information, references, etc.

Introduction: I would like to have more information about how theses techniques have been tested and used to generate land cover maps.

Response:

Thank you for your constructive comments. We have added several sentences to describe how these methods can be used and tested for land cover mapping. For example, see lines 57-58, where we review several applications of common SRM methods to generate land cover maps. See also lines 100-108, which gives examples of land cover mapping based on CNN.

Methods: You write absolute sentences without any reference supporting your writing. For example, line 133; "The convolution layer is the most basic and inportant module in CNN", is it your opinion or are there evidences to write it?. Review your document.

Response:

Thank you for your careful review. We have added them with suitable reference, which in shown in Line 146, 154, 162, 166.

Figures are difficult to read, for example, in Fig 1 and 2 text is too small.

Response:

We have revised the Figure 1 and Figure 2 and increased font size.

3.1. Datasets: Please provide references as web pages for datasets used. Moreover, can you provide information about image characteristics. I mean, sensor, pixel size...

Response:

We have added the requested information, see lines a web reference for this sentence, which is shown in lines 220, 221, 222. We could not add information about the sensor because that was not available.

Results: I appreciate if you show add producer accuracy. It would be interesting a deep analyze between classes. What is background class?

Response:

Thank you for your constructive advice. We have added the analysis for PA between different classes, which are shown in lines 313-316 and 349-351. The background class represent the area with impervious, bareland and cars, which is shown on the dataset webpage.

You have worked with just only 3 classes. What is the effect of number of classes? A real Land cover maps is not only 3 classes.

Response:

Thank you for this comment. The reason for selecting three classes is based on the band number of input image and the landscape characteristic and we have added several sentences to describe it, which are shown in Line 224-226 and 236-239.

Additionally, we text on increasing the number of classes to the Discussion in future work from real Landsat or Sentinel 2 images, see lines part about future work, which is how generate finer land cover map from real Landsat or Sentinel 2 image. They are shown in lines 446-451.

Dicussion section: You do not discuss anything at all, you just show results. Please rewrite it, link with previous results from other researcher results.

Response:

We have rewritten the Discussion part, and now make a comparison with results from other researches.

Conclusion: Please, rewrite it. It is a very short summary of your result section.

Response:

We have rewritten the Conclusion.

Finally, what about future research?

Response:

We have added a part in the Discussion part about future work. We think that the main future work on CNN-based SRM method can be concluded into three aspects: networks selection, training sample collections, and multi-scale integration.

How can your methods can be incorporated in real flowchart to generate land cover maps?

Response:

We address this now in the Discussion when we mention potential future work. See lines 436-442.

Yo have used fine spatial resolution with 9 and 5 cm, what pixel size effect?. I mean, what is your opinion about this methodology if a used Landsat or Sentinel 2 images to generate a land cover map of a country? 

Response:

We address this now in the Discussion. See lines 444-451 where we describe how  may be used to generate a national or world land cover map.

Round 2

Reviewer 3 Report

Authors have taken into account my suggestions and manuscript is aceppted in present form